# Structural phase transition in NH$_4$F under extreme pressure conditions
Umbertoluca Ranieri [1,2], Christophe Bellin[3], Lewis J. Conway[4,5], Richard Gaal[6], John S. Loveday[2], Andreas Hermann [2], Abhay Shukla [3,7] ✉ & Livia E. Bove [1,3,6,7] ✉

Ammonium fluoride (NH$_4$F) exhibits a variety of crystalline phases depending on temperature and pressure. By employing Raman spectroscopy and synchrotron X-ray diffraction beyond megabar pressures (up to 140 GPa), we have here observed a novel dense solid phase of NH$_4$F, characterised by the tetragonal *P4/nmm* structure also observed in other ammonium halides under less extreme pressure conditions, typically a few GPa. Using detailed ab-initio calculations and reevaluating earlier theoretical models pertaining to other ammonium halides, we examine the microscopic mechanisms underlying the transition from the low-pressure cubic phase (*P-43m*) to the newly identified high-pressure tetragonal phase (*P4/nmm*). Notably, NH$_4$F exhibits distinctive properties compared to its counterparts, resulting in a significantly broader pressure range over which this transition unfolds, facilitating the identification of its various stages. Our analysis points to a synergistic interplay driving the transition to the *P4/nmm* phase, which we name phase VIII. At intermediate pressures (around 40 GPa), a displacive transition of fluorine ions initiates a tetragonal distortion of the cubic phase. Subsequently, at higher pressures (around 115 GPa), every second ammonium ion undergoes a rotational shift, adopting an anti-tetrahedral arrangement. This coupled effect orchestrates the transition process, leading to the formation of the tetragonal phase.

Ammonium fluoride (NH$_4$F) is considered the odd man out among ionic ammonium halides, being isostructural to hexagonal ice (Ih) at ambient conditions and having a similar low-pressure phase diagram. In a recent work[1] (Bellin et al.), some of us have shown that around 10 GPa a disorder-order transition, reminiscent of the ice VII to ice VIII transition, is found in NH$_4$F at low temperature. With further increase in pressure, chemical bonding and structural phases in ammonium fluoride resemble those in other ammonium halides. In this study, we extend our investigations using Raman spectroscopy and X-ray diffraction techniques applied to NH$_4$F, pushing the boundaries beyond the megabar regime. Our observations reveal a notable cubic to tetragonal phase transition, akin to those observed in other halides but occurring at significantly lower pressures. This transition, from the space group *P-43m* (No. 215) to the space group *P4/nmm* (No. 129), has been extensively studied both experimentally and theoretically.

The *P-43m* phase, which is known as phase III of NH$_4$F, adopts an ordered CsCl-like cubic structure. This order manifests in the consistent rotational orientation of NH$_4$ tetrahedra throughout the crystal lattice. Phase III originates from a disordered CsCl-like structure existing at lower pressures or higher temperatures, where NH$_4$ tetrahedra lack such ordered orientation. In a recent computational study[2] (Conway et al.), some of us explored NH$_4$F into the multi-megabar regime and predicted a transition from phase III to a series of denser structures above 80 GPa. These structures necessitate significant rearrangement of both the fluorine and NH$_4$ sub-lattices. Furthermore, ref. 2 found that the *P4/nmm* structure is more stable than phase III beyond 110 GPa.

The rotational orientation of the ammonium tetrahedra is one of the two key parameters, the other one being the position of the halide ion, that plays a major role in the cubic to tetragonal transition. This *P-43m* to *P4/nmm* transition takes place at about 11 GPa in NH$_4$Cl[3], 6.5 GPa in NH$_4$Br[4] and 5 GPa in NH$_4$I[5], the transition pressure decreasing with increasing size of the halide ion and increasing strength of the hydrogen bond in the material. It involves an anti-tetrahedral type ordering of the NH$_4$ tetrahedra as well as that of the halide ion position within the unit cell. Yamada et al.[6]

[1]Dipartimento di Fisica, Università di Roma La Sapienza, Rome, Italy. [2]Centre for Science at Extreme Conditions and School of Physics and Astronomy, University of Edinburgh, Edinburgh, UK. [3]Institut de Minéralogie, de Physique des Matériaux et de Cosmochimie, Sorbonne Université, CNRS UMR 7590, Paris, France. [4]Department of Materials Science and Metallurgy, University of Cambridge, Cambridge, UK. [5]Advanced Institute for Materials Research, Tohoku University, Sendai, Japan. [6]Laboratory of Quantum Magnetism, Institute of Physics, École Polytechnique Fédérale de Lausanne, Lausanne, Switzerland. [7]These authors jointly supervised this work: Abhay Shukla, Livia E. Bove. ✉e-mail: abhay.shukla@sorbonne-universite.fr; livia.bove@sorbonne-universite.fr

remarked that the two possible relative orientations of adjacent ammonium tetrahedra can be described by a phenomenological Ising pseudospin variable, while the halide ion displacement corresponds to the freezing of a zone boundary acoustic phonon in these systems. They developed a model microscopic Hamiltonian to describe the spin-phonon coupled system which includes a direct electrostatic coupling between tetrahedra that dictates their *tetrahedral* alignment while the indirect phonon mediated coupling is responsible for the *anti-tetrahedral* order with adjacent tetrahedra rotated by 90°. By including free energy considerations and investigating thermodynamic stability, this model was remarkably successful in explaining the different phase diagrams of $NH_4Cl$, $NH_4Br$ and $NH_4I$[7,8].

In this work, we use high-pressure Raman spectroscopy and X-ray diffraction (XRD) to investigate the structural phases of $NH_4F$ up to 140 GPa at room temperature. We identify the existence of the *P4/nmm* phase at very high pressures, which is here named phase VIII of $NH_4F$ (note that a recent study of $NH_4F$'s low-pressure polymorphs suggests that phases V to VII assigned in the 1960's[8] might not be required[9]). Our findings confirm that with increasing pressure and hydrogen bond strength, $NH_4F$ more closely resembles the other halides of the family. However, this new phase appears at a pressure that is an order of magnitude higher than in the other halides and the transition from the cubic III phase to the tetrahedral *P4/nmm* structure extends over a pressure range of almost 80 GPa. This extended pressure range allows for better insights into the transition mechanism and the possibility to investigate the role of the tetragonal distortion of the cubic phase III (identified by Bellin et al. as phase IIIt[1]) of $NH_4F$, which has been observed at pressures as low as 10 GPa, at low temperature. We observe here that at room temperature, the IIIt phase appears in the phase diagram above 40 GPa and is the precursor of phase VIII. We assign space group *P-4m2* (No. 115) to the IIIt phase.

These observations not only support the hypotheses of earlier phenomenological spin-phonon models applied to other ammonium halides but also highlight the clear divergence of $NH_4F$ from water ice as pressure intensifies. For instance, unlike the proton ordering transition from ice VII to ice VIII, which occurs solely by lowering the temperature at constant pressure in ice, our findings indicate a different thermodynamic trajectory for $NH_4F$. In ice, the hydrogen bond symmetry changes drastically with pressure, leading to the formation of the non-molecular ice X phase at around 60 GPa.

To deepen our understanding of the dynamics and energetics underlying the phase transition in $NH_4F$, we conducted extensive ab initio calculations encompassing total energy, structural evolutions and Raman spectroscopy, alongside exploring energy pathways between different structures. Our simulations not only confirm the existence of both phases III and VIII but also accurately replicate their structures and Raman spectra, demonstrating a high level of agreement.

Moreover, our simulations suggest that the transition to phase VIII is driven by the combined effect of a displacive transition of fluorine ions—already evident in the intermediate IIIt phase—and the subsequent rotation of ammonium ions in alternate 'antiferromagnetic' units at higher pressure. This rotational motion facilitates further displacement of the fluorine positions, contributing to the transition process.

## Results

Several samples were loaded in diamond anvil cells for this study (see Methods and Supplementary Fig. 1). The temperature-pressure phase diagram of $NH_4F$ is reported in Fig. 1. It includes literature knowledge as well as observations made in this work. The phase transition boundary at 115 GPa at room temperature is obviously affected by a ±5 GPa incertitude, since measurements were made at reasonably large intervals, given the extended pressure range investigated.

### X-ray diffraction measurements and related computational results

Information about the structural transitions under high pressure in $NH_4F$ can be obtained through X-ray diffraction. In Fig. 2, we show the results of

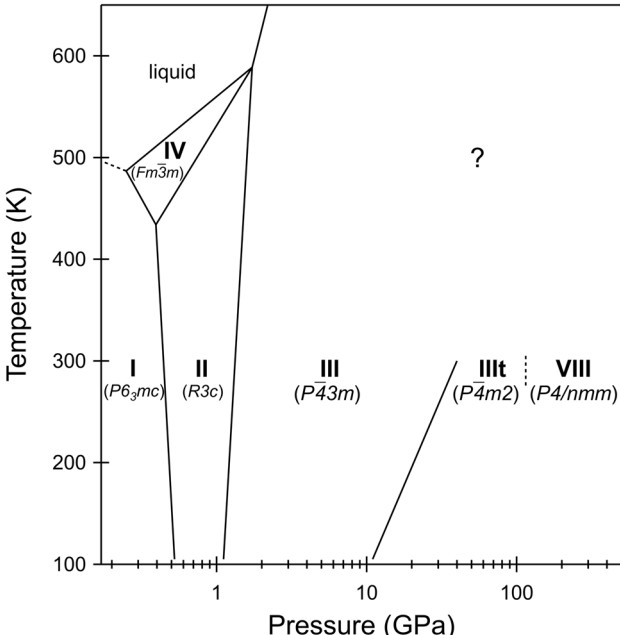

**Fig. 1 | Phase diagram of NH₄F.** Schematic phase diagram of ammonium fluoride including the phase transition from the tetragonal phase IIIt to the tetragonal phase VIII found in this work. Space groups are given in parentheses.

our experiment for data collected upon compression between 48 and 133 GPa on the ID15b beamline at the ESRF. In Bellin et al.'s earlier work[1], X-ray diffraction data were limited to 25 GPa at room temperature and were indexed using a cubic cell (phase III).

The present measurements reveal that the main, i.e. reflection (1 1 0) of phase III, diffraction peak (2θ ~ 11.5° at 47.6 GPa) becomes increasingly asymmetric with increasing pressure but cannot be unequivocally resolved into two separate Bragg reflections over the pressure range below 115 GPa (red patterns in Fig. 2b, c). This is the stability region of the IIIt phase, as further explained below and the asymmetry arises from the small tetragonal distortion. A comparison of the calculated diffraction patterns of the phases III and IIIt at 50 GPa is reported in Supplementary Fig. 2. Reflection (1 0 0) of the Re gasket is visible in all patterns, meaning that reflection (1 0 1) must be also contributing to the main diffraction signal at 11.5–12°, although it is much weaker than the $NH_4F$ signal (see Supplementary Fig. 3). In some patterns (mostly at 47.6 GPa), the reflection (1 1 1) of gold is also visible.

Above around 115 GPa, the stability region of phase VIII (blue patterns in Fig. 2b, c), the main diffraction signal can be clearly resolved into two reflections: (2 0 0) and (1 1 1), which are indeed predicted to be the most intense (see calculated pattern in Fig. 2a). The picture emerging from X-ray diffraction is a continuous transition from the III to the IIIt phase and from the IIIt to the VIII phase upon compression.

The two main reflections can be fitted using two Gaussians throughout the investigated pressure range and no major discontinuities in their positions are observed. Over the pressure range above 115 GPa, for which the splitting is clearly resolved, the ratio in the widths of the two Gaussians was found to be pressure independent and close to a value of 1.5. The same ratio was imposed in the fits over the pressure range below 115 GPa. Tetragonal cell dimensions and volumes, which were extracted from the positions of the two Gaussians, are reported as a function of pressure in Figs. 2d, e. They agree very well with those we calculated for phase VIII with the *P4/nmm* structure. Phase IIIt is not found to be stable in our calculations (it reverts to the cubic phase III) so the simulated volumes for phase III are reported instead. The structural details of the IIIt phase cannot be determined reliably from our measurements due to the small number of reflections available and pronounced preferential orientation of sample grains. However, phase IIIt is only a slight structural distortion of phase III, and an intermediate structure

https://doi.org/10.1038/s42004-024-01309-w                                                                                      **Article**

between the phases III and VIII. Both space groups *P-42m*, suggested in Bellin et al.'s earlier work[1], and *P-4m2* are compatible with our diffraction data of phase IIIt. They only differ in the fact that space group *P-4m2* allows for a displacement of F ions and thus better accounts for the observed tetrahedral distortion. The Raman spectroscopy measurements presented below are compatible with the *P-4m2* structure but not with *P-42m*, supporting our choice of *P-4m2*. *P-4m2* has the same larger ($\sqrt{2} \times \sqrt{2} \times 1$) tetragonal cell as *P4/nmm* and the splitting is also assigned to reflections (2 0 0) and (1 1 1). In the *P4/nmm* structure (phase VIII), a further displacement of the F ion positions is induced due to an anti-tetrahedral ordering of the $NH_4$ tetrahedra. The structural details of all the phases discussed in this manuscript are reported in Supplementary Tables 1 and 2, and as Supplementary Data 1–3.

## Raman spectroscopy measurements and related computational results

Raman spectroscopy is a commonly employed tool to investigate structural changes as vibrational modes are very sensitive to the local molecular environment. In Fig. 3 we show Raman spectra as a function of pressure for $NH_4F$ at room temperature up to 140 GPa. The pressure range below 40 GPa was studied closely in Bellin et al.'s earlier work[1]. Here we focus on the high-pressure range from 40 to 140 GPa. The three panels of Fig. 3 show respectively (a) the extended lattice mode, (b) the local bending and (c) the stretching modes. The lattice mode hardens above 40 GPa with a significant broadening till 81 GPa but no sharp evolution indicating a structural transition is observed. From 91 to 109 GPa, the lattice mode is not detectable and this will be discussed later.

The bending mode also broadens above 40 GPa and evolves continuously till 114 GPa, above which a jump in the bending frequency and a clear doubling of the mode is observed. The splitting associated with the tetragonal distortion mentioned above was detected in these two modes in Bellin et al.'s earlier work[1] at low temperature. Bellin et al. had previously found that the onset temperature of this splitting increased with increasing pressure. The broadening observed at room temperature in both lattice and bending modes above 40 GPa is compatible with this observation. Finally, the stretching mode also undergoes a first slight change at 40 GPa both in frequency and in relative intensity of the modes, followed by a continuous evolution with pressure up to 114 GPa. At 118 GPa and above, a clear change in the frequency, number and intensity of the modes is observed. Concurrently, the lattice mode now displays two peaks, red-shifted, sharper and with lower intensity than the single broad peak at lower pressure. The bending and stretching modes also evolve into two well-separated peaks at higher frequency. The precise frequency of the modes was obtained by fitting the peaks evolution as a function of pressure with a combination of Lorentzian and Gaussian lineshapes and is reported in Fig. 4.

The upper panels of Fig. 4 show the experimental Raman frequencies and their evolution with pressure while the lower panels show our simulation results. The area of the dots corresponds to the intensity of the Raman peak while the colours identify phases. While we reproduce phases III and VIII, the IIIt phase is not stable in ground state ab initio calculations, as mentioned earlier. The simulations of the Raman spectra of phase IIIt were thus performed constraining the cell dimensions to the experimental values. The F displacement, meanwhile, was set as follows: using the experimental unit cell at 114 GPa, we set up a series of F-displacements of varying amplitudes, optimised all other degrees of freedom and then simulated the Raman spectra. We compared these series of simulated spectra to the experimental spectrum (see Supplementary Fig. 4), which allowed us to establish the displacement amplitude at 114 GPa, where the F atoms are at position (0, ½, z) with z = 0.0405. We then assumed a linear dependence of the displacement on the c/a ratio, starting from z = 0 at 40 GPa, the onset of phase IIIt, as shown in Supplementary Fig. 5. Finally we simulated Raman spectra over the entire stability range of phase IIIt, again optimising all atomic degrees of freedom aside from the F displacements. Our simulated Raman frequencies agree remarkably well with the experimental results, including the blue shift of modes in the IIIt phase with respect to phase III.

The substantial overall agreement between experiment and simulations bears out our preliminary conclusions. The subtle room-temperature distortion (phase IIIt) above 40 GPa, as seen by the splitting of lattice and bending modes, and the new tetragonal phase (phase VIII) above 115 GPa are well reproduced by calculations. Up to five stretching modes are seen in the measured spectra over the pressure stability range of the IIIt phase. This agrees with the simulated Raman of the *P-4m2* structure, which indeed contains extra stretching modes as compared to the simulated Raman of the *P-42m* phase. Our calculated Raman frequencies and intensities for *P-42m* are reported in Supplementary Fig. 6, while the mode characters for the phases III, IIIt and VIII are tabulated in Supplementary Tables 3, 4 and 5, respectively. Some discrepancies between experiment and simulations are discussed below. A mode at about 650 $cm^{-1}$ in the simulations is not detected in this experimental data. However, in Bellin et al.'s earlier work[1] this mode was measured and attributed to an ammonium ion librational mode ($v_6$), also seen in other halides as a combination mode or an overtone. Another mode at about 2100 $cm^{-1}$ experimentally seen in the low-pressure region is not reproduced by our simulations. This mode was identified in Bellin et al.'s earlier work[1] as a combination mode ($v_4 + v_6$) also observed in the other ammonium halides. The absence of an experimental lattice mode in the region 91–109 GPa is very probably due to the corresponding small simulated intensity for the IIIt phase but also defective martensitic regions in this pressure range where IIIt in some regions transforms microscopically to VIII. Finally, the computational study[2] by Conway et al. predicted two other high-pressure phases to be more stable than the *P4/nmm* structure. Our X-ray diffraction and Raman measurements rule out their formation. Supplementary Fig. 7 reports the calculated diffraction patterns, and Supplementary Figs. 8 and 9 report the calculated Raman of the two phases.

## Computational insights into the transition mechanism

As noted earlier, while the *P-43m* and *P4/nmm* phases are common to the whole ammonium halide family, for $NH_4Cl$, $NH_4Br$ and $NH_4I$ the transition between the two structures is direct and takes place close to 10 GPa. In $NH_4F$, the *P4/nmm* phase VIII sets in only at around 115 GPa, which agrees very well with Conway et al.'s previous predictions[2]. The *P-43m* cubic phase III of $NH_4F$ clearly exists at low pressure. Above 40 GPa and at room temperature, the tetragonally distorted phase (IIIt) sets in. This tetragonal distortion is already visible at 10 GPa below 100 K[1]. The two parameters that control the transition are the $NH_4F$ tetrahedron rotation[10,11] which switches from a tetrahedral (phases III and IIIt) to anti-tetrahedral long-range order (phase VIII) and the tetragonal distortion which in the other halides has been identified as arising from a frozen zone boundary phonon which involves exactly the same halide ion displacements[12,13] (Fig. 5a). In $NH_4F$, we thus have a range extending from 40 GPa to 115 GPa in which the stages of the phase transition can be investigated. The overall result of the III–VIII transition is a denser structure but also a diversification of local environments, as visualised by the heavy atom separation histograms in Fig. 5b.

In Fig. 5c, we explore the 2D potential enthalpy surface along the two symmetry-lowering parameters by evaluating the relative enthalpies of structures that are intermediate between those of the III and VIII phases, characterised by specific partially rotated $NH_4$ groups as well as F ion displacements. The x-axes correspond to $NH_4$ rotation (in degrees), the y-axes to F ion displacement and tetragonal distortion (relative to their magnitudes in phase VIII). The colour scale pertains to relative energetics, with black to blue to white reflecting increasing enthalpy. At moderate pressures (see 50 GPa results), the F ion displacement in phase III is easily facilitated (small energy cost along the y-axis) while $NH_4F$ rotation is strongly impeded (high barrier in the almost vertical white region), preventing rotations away from non-zero values. This could explain why the IIIt phase evolves first as a tetragonal distortion of the cubic phase, driven by F ion displacement with the 'freezing' of the corresponding zone boundary phonon. At 100 and 150 GPa, the barrier towards the anti-tetrahedral rotation is lower and there is increased coupling between both degrees of freedom.

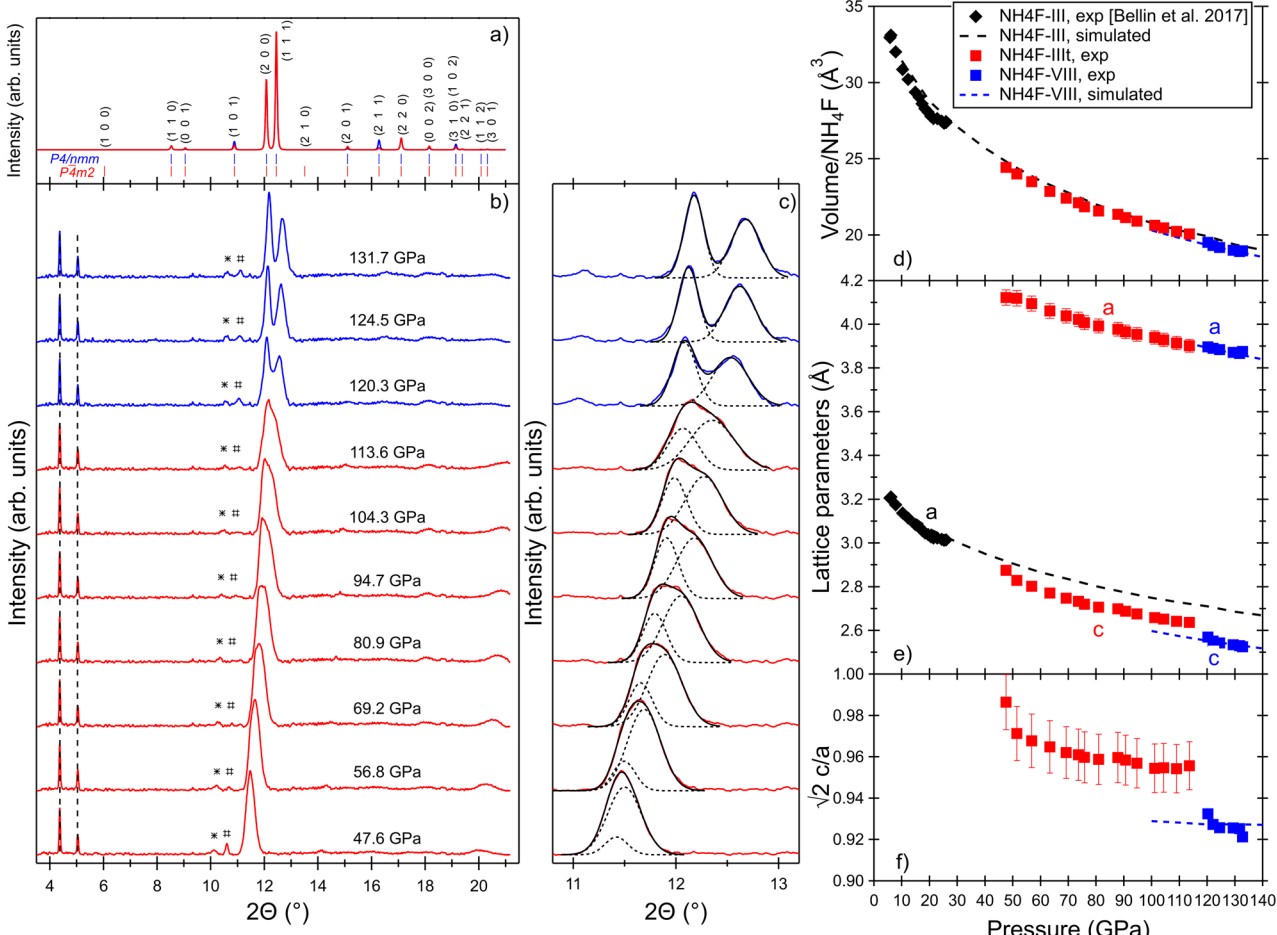

**Fig. 2 | X-ray diffraction patterns and their analysis. a** Calculated X-ray diffraction patterns for perfect powders of NH$_4$F-IIIt (space group *P-4m2*) and NH$_4$F-VIII (space group *P4/nmm*) at 115 GPa (*a* = 3.9 Å and *c* = 2.6 Å). Tick marks indicate the positions of the Bragg reflections and Miller indices are also reported.
**b** Representative X-ray diffraction patterns of the sample at the indicated pressures (λ = 0.4101 Å). The symbols * and # indicate a Re (gasket material) and a Au (pressure standard) peak, respectively. The vertical lines indicate two peaks which do not emanate from the sample (their positions do not shift with pressure). **c** Best fits of the most intense diffraction signal using two Gaussians. **d** Unit cell volume per number of NH$_4$F molecules in the cell. **e** Cell dimensions. **f** √2c/a ratio. Error bars correspond to one standard deviation. Legend of (**d**) also applies to (**e**, **f**). Symbols represent experimental data, dashed lines represent DFT calculations. Data of ref. [1] are also reported.

However, this simple calculation does not optimise internal degrees of freedom since atomic positions and lattice vectors are fixed. In reality, the transition will be along a complex reaction coordinate that combines NH$_4$ rotation, ion displacements and localised atomic motion to avoid large energy barriers. To partially account for this complexity, we performed nudged elastic band (NEB) calculations[14], set up initially along the orange pathways shown in Fig. 5c. Within this scheme a sequence of 16 structural 'images' that interpolate between the initial (NH$_4$F-III) and final (NH$_4$F-VIII) states are optimised, while kept apart by harmonic springs. Ideally, this will lead to a sequence of structures that follow the energetically least costly transition path, and therefore give a more realistic transition barrier. The enthalpies along the NEB paths relative to phase III are shown in Fig. 5d, and the resulting transition barriers are 0.34/0.25/0.15 eV/NH$_4$F at 50/100/ 150 GPa. Not only does phase VIII become energetically more stable than phase III at higher pressures, but the barrier towards that transition becomes significantly lower, thereby adhering to the Bell-Evans-Polanyi principle. The barrier and the transition state depend on our initial assumption of the reaction pathway; it is known from other systems that more complex reaction pathways can result in significantly lower energy barriers[15,16]. The calculated Raman-active modes at the saddle points at 50, 100 and 150 GPa are reported in the Supplementary (Supplementary Figs. 10 and 11), along with the structural details (Supplementary Table 6 and Supplementary Data 4–6).

## Discussion and outlook

To summarise, we have combined X-ray diffraction and Raman spectroscopy measurements beyond megabar pressure with computer simulations for identifying the *P4/nmm* phase in NH$_4$F above 115 GPa, a phase found in other ammonium halides at much lower pressures. The emerging picture agrees with the earlier Ising spin-phonon theory in heavier halides, which treats the rotational orientation as a pseudo-spin and examines both direct (spin/rotation) and indirect (phonon/displacement) interaction between NH$_4$ tetrahedra to explain the phase transition. The *P4/nmm* tetragonal phase, which combines the rotation of NH$_4$ tetrahedra and the displacement of F ions along the tetragonal axis, both in an anti-tetrahedral arrangement, is possible in NH$_4$F at 115 GPa, despite the energy cost of the rotation because of the stabilising nature of the coupled displacement of the F ions.

In an earlier work by some of us[1], Bellin et al. had identified a tetragonal distortion in NH$_4$F as seen through the splitting of Raman lines at low pressure and temperature, and identified it with a new phase which was labelled with the most symmetric corresponding space group *P-42m*. However, we had not yet detected a physical mechanism for this distortion. In this work, by largely extending the pressure regime to beyond the megabar, we uncover the microscopic mechanism of a phase transition which is found in ammonium halides and has been studied in the past in other members of this family[9]. This phase transition, which takes place in NH$_4$Br as a function of temperature, has been explained in the past as an example of a coupled

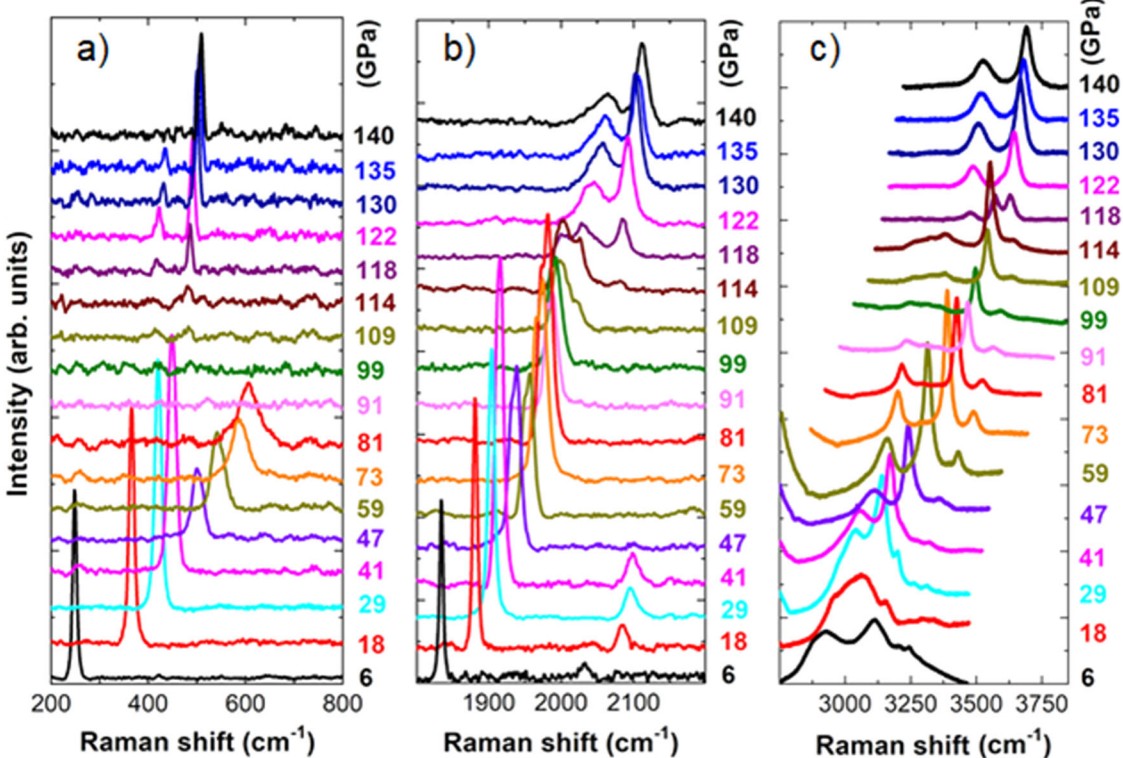

**Fig. 3 | Raman spectroscopy data.** Raman spectra measured upon compression at pressures from 6 to 140 GPa: (**a**) lattice mode; (**b**) $NH_4^+$ bending region and (**c**) N-H stretching region. All measured pressure points are reported.

displacive and order-disorder phase transition[8,11] involving simultaneous displacements of the halide ion (phonon freezing, displacive) and proton ordering through rotation of $NH_4$ tetrahedra (pseudospin, order-disorder).

In this work, we identify the space group of the tetragonally distorted intermediate phase as *P-4m2*. This space group allows for displacements of the F ions along the c-axis, which is needed to explain observed additional Raman peaks. In $NH_4F$, the transition takes place over a broad pressure range, and the displacive and order-disorder components of the phase transition are effectively decoupled. Bellin et al. showed in their earlier work[1] that a small tetragonal distortion is visible at low temperature for pressures as low as 10 GPa. In this work, we show that above 40 GPa, the tetragonal phase is well established as a result of phonon condensation and F ion displacement which increases with pressure. Between 80 and 115 GPa, the rotation of the aligned $NH_4$ tetrahedra becomes energetically possible, notably with the increasing displacement of the F ions, giving rise to disorder and the suppression of the lattice mode in the Raman spectra. Above 115 GPa, the $NH_4$ tetrahedra settle into an alternate anti-tetrahedral orientation which accompanies the F ion displacement and establishes the *P4/nmm* structure with reappearance of the lattice mode.

Previous electronic structure calculations[2] by some of us successfully predicted that the *P4/nmm* phase should become more stable than phase III (*P-43m*) above 110 GPa. Here, we also succeed in simulating Raman spectra of phase III, phase IIIt, as well as phase VIII (*P4/nmm*) in quantitative agreement with experiments. However, Conway et al.'s original calculations[2] predicted different, more stable high-pressure phases and failed to identify a stable tetragonally distorted structure of phase III. The former is likely due to reaction kinetics: as discussed above, there is a viable low-barrier pathway from phase III to phase VIII, while other high-pressure phases predicted by Conway et al.[2] require major rearrangements of the ionic sublattices. This issue is not unique to $NH_4F$: another ionic hydrogen-bonded system where a predicted high-pressure phase is only seen as minority compound in experiment is ammonia monohydrate, where topological frustration between the ionic sublattices might prevent the high-pressure phase from forming on the experimental time scale[17–19].

The lack of a stable tetragonal phase IIIt in calculations is a different issue, and suggests more theoretical work on this compound would be welcome; a self-consistent de-stabilisation of the zone-boundary F displacement phonon at finite temperatures, coupled to a tetragonal lattice distortion, would deliver the desired result. Nuclear quantum effects or anharmonic renormalisation of phonon frequencies can stabilise crystal structures that are found to be dynamically unstable if nuclei are assumed to be classical point particles in purely harmonic potentials[20–22]. One example is the stabilisation of high-symmetry cubic $LaH_{10}$, driven by symmetrisation of the hydrogenic sublattice[21,23]. The IIIt phase is quite different, however: it is a *low-symmetry* structure, driven by displacement of the *fluorine* sublattice, while hydrogens remain in place as in phase III. It is therefore not immediately clear how nuclear quantum effects would stabilise the IIIt phase, while the potential energy surface along the F displacement (see Fig. 5c) does not suggest strong anharmonicity either. Nuclear quantum effects for hydrogen could result in $NH_4$ cations being quantum or (at elevated temperature) classical rotors. That would mean that $NH_4$ cations, upon compression, evolve from fixed orientation (in phase III) to plastic/rotational states (in phase IIIt) and back to fixed orientation (in phase VIII). There is no indication for this in the experimental data, and no precedence in the literature either: plastic or rotational states in water, ammonia, or their mixtures, are all arrived at upon heating, not compression[24–27].

$NH_4F$ is known to be a typical hydrogen-bonded structure at ambient and moderate pressures. Here we confirmed that it increasingly resembles other ionic ammonium halides as pressure increases. We also show that it allows for the microscopic understanding of a complex phase transition common to ammonium halides by decoupling the intervening mechanisms over a wide pressure range.

## Methods
### Experimental
Three symmetric screw diamond anvil cells (DACs) equipped with 100 µm culet diamond anvils were prepared. Rhenium was used as the gasket material and its initial thickness of 250 µm was indented to ~25 µm.

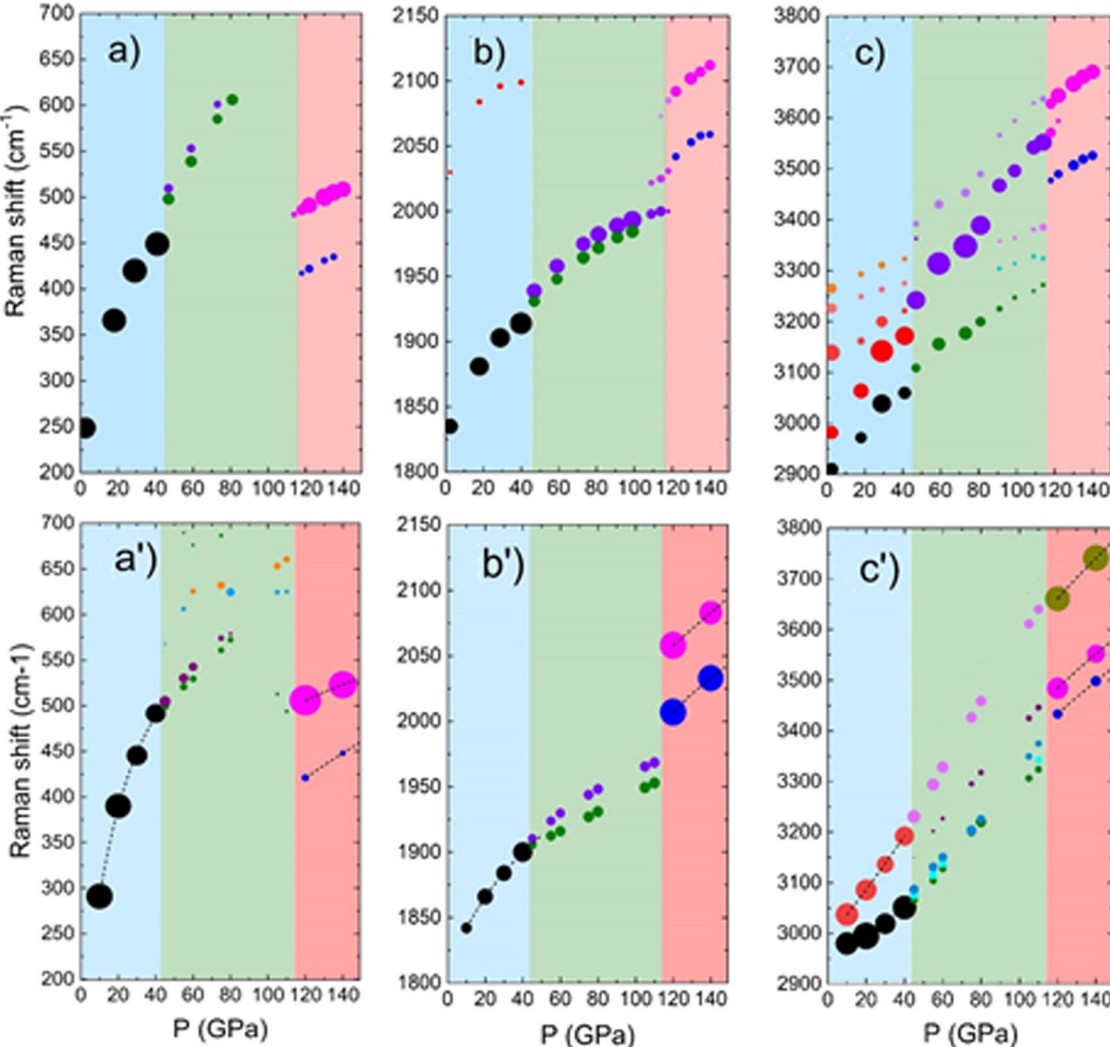

**Fig. 4 | Variation of the Raman shift as a function of the pressure.** Top panels: measured (**a**) lattice mode, (**b**) bending region and (**c**) stretching region. The shaded areas highlight the III to IIIt phase transition then the IIIt to VIII phase transition at higher pressure. Lower panels: theoretical (**a'**) lattice mode, (**b'**) bending region and (**c'**) stretching region. The size of the dots is given by the intensity of the Raman peaks.

The sample cavities of about 50 μm in diameter were laser drilled at the centre of the indentations. To avoid any presence of water, the NH₄F polycrystalline samples of 99.8% purity from Sigma Aldrich were dried at 400 K for 30 min before being loaded in the DACs under Argon atmosphere in a glovebox. Neon or Argon gas at a pressure of 0.2 GPa was finally loaded to serve as a pressure transmitting medium (PTM).

DAC#1 and DAC#2, which had respectively Neon and Argon as PTM, were employed for Raman spectroscopy measurements exclusively. Both DACs were compressed stepwise in pressure up to 140 GPa and Raman spectra were collected at every pressure point. The Raman data reported are those collected on DAC#1. The data collected on DAC#2 confirmed the phase transition occurring at ~115 GPa. In addition to the measurements taken upon compression, DAC#2 was also decompressed from 140 GPa down to ambient pressure while collecting Raman spectra, confirming that the phase transition at ~115 GPa is reversible. DAC#3, which was loaded with Neon gas as PTM, was measured by synchrotron powder XRD upon compression from 48 to 133 GPa (at the Beamline ID15b, see below). This compression was performed using the inflation of a membrane which was in contact with the DAC inside a metallic cylinder. After these measurements and the release of the membrane pressure, sample pressure had dropped to 110 GPa. Raman spectra were then taken upon compression from 110 to 135 GPa. These additional Raman measurements (not shown here) reproduced the observations done on DAC#1 and DAC#2. Subsequently, powder

XRD data were collected at 135 GPa (at the Beamline ID27, see below). This was done to confirm the correspondence between the 115 GPa phase transition seen by Raman and that seen by XRD and exclude a significant effect of the compression rate (which was faster for the synchrotron XRD experiment than for the laboratory Raman measurements). Visual inspection shows a non-metallic sample at all pressures. Examples of photomicrographs of the three samples are reported in Supplementary Fig. 1.

The XRD data reported were collected at the High-Pressure Diffraction Beamline ID15b ($\lambda = 0.4101$ Å) of the European Synchrotron Research Facility (ESRF) using an Eiger2 X 9 M CdTe flat panel detector. The X-ray beam spot size was $6 \times 6$ μm. Complementary XRD data were collected at the Beamline ID27 ($\lambda = 0.3738$ Å) of the ESRF, as mentioned above. The 2D diffraction images were processed to mask the Bragg peaks of the diamond anvils then integrated into one-dimensional patterns, and a smooth polynomial background was subtracted with the programme Dioptas[28]. Reference diffraction patterns were recorded on the Re gasket at selected pressures. The crystal structures were visualised with the programme Vesta[29]. In the XRD measurements, pressure was determined using the equation of state of Au[30].

Two different Raman setups were employed. DAC#1 was measured using a Jobin-Yvon HR-460 single monochromator spectrometer in a backscattering configuration with a 1500 grooves/mm grating and equipped with an Andor CCD camera. A 514.5 nm wavelength radiation from an Ar laser

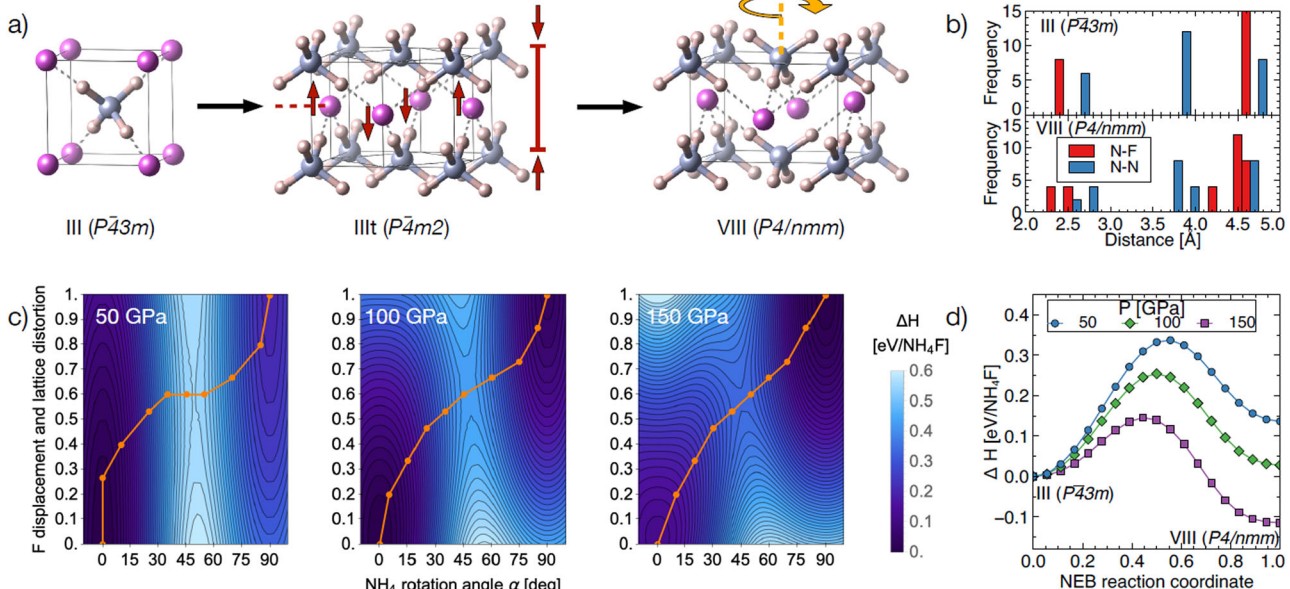

**Fig. 5 | Structural differences and transition path. a** Transition pattern from the NH$_4$F-III (space group $P\bar{4}3m$) to the NH$_4$F-VIII (space group $P4/nmm$) phase, including the intermediate NH$_4$F-IIIt (space group $P\bar{4}m2$) phase. The pattern shows the lattice tetragonal distortion linked to the F displacement in phase IIIt (red arrows), and the rotation of every second NH$_4$ cation (anti-tetrahedral order) accompanied by further displacement of F ions in phase VIII (orange arrow). **b** Pair distribution functions for the heavy atom lattices for NH$_4$F-III and NH$_4$F-VIII at 100 GPa. **c** 2D potential energy surfaces in NH$_4$F, linking NH$_4$F-III at (0,0) to NH$_4$F-VIII at (90,1.0), where the x coordinate is the NH$_4$ tetrahedron rotation in degrees and the y coordinate is the F ion displacement and tetragonal distortion relative to phase VIII. From left to right the calculations are for 50/100/150 GPa. **d** Relative enthalpies for nudged elastic band calculations from phase III to phase VIII, initialised along the orange paths shown in (**c**).

was focused into a 2 × 2 μm spot by a long-working distance Mitutoyo x20 objective. The power of the laser measured directly on the DAC was kept below 2 mW to avoid any photoinduced transformation of the sample. For DAC#2 and DAC#3, measurements were performed using an HR800 commercial confocal Raman spectrometer equipped with Cobolt 532-nm 2-W DPSS laser. We used a long working distance 50× Mitutoyo objective. Spectra were collected both with low (600 grooves/mm grating) spectral resolution for very broad spectral features in the 500–4500 cm$^{-1}$ range and high (1800 grooves/mm grating) spectral resolution for more detailed features in the 100–1200 cm$^{-1}$ range. A smooth background was subtracted from the Raman spectra before fitting the data. Also, reference Raman spectra were recorded at each pressure on the gasket close to the sample chamber. Pressure was determined by the shift of the R1 ruby fluorescence line[31] and by the pressure shift of the first order Raman band of the diamond anvils for the highest pressures[32].

All measurements were performed at room temperature.

## Computational

All calculations were performed using density functional theory in conjunction with plane wave basis sets as implemented in the CASTEP and VASP software packages[33,34]. Electronic exchange-correlation effects were modelled using the Perdew-Burke-Ernzerhof functional[35]. CASTEP calculations used Brillouin zone sampling grids with spacing no larger than $2\pi*0.04\,\text{Å}^{-1}$, combined with ultrasoft pseudopotentials generated 'on-the-fly' and plane wave cutoff energies of 1000 eV for equations-of-state, and with norm-conserving pseudopotentials generated 'on-the-fly' and plane wave cutoff energies of 700 eV for simulated Raman spectra. VASP calculations used fixed zone-centred (6,6,8) Brillouin zone sampling grids combined with 'hard' projector augmented wave datasets[36] and plane wave cutoff energies of 1000 eV for potential energy surface and nudged elastic band calculations.

## Data availability

The data are available from the corresponding authors upon request.

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

## Acknowledgements

We acknowledge the European Synchrotron Radiation Facility for provision of synchrotron radiation facilities, and we would like to thank Michael Hanfland and Mohamed Mezouar for assistance in using the beamlines ID15b and ID27. We are grateful for computational support from the UK national high performance computing service, ARCHER2, for which access was obtained via the UKCP consortium and funded by EPSRC grant ref EP/X035891/1. L.E.B. acknowledges funding through the ANR-23-CE30-0034 EXOTIC-ICE, the Swiss National Fund (FNS) grant EXOTIC-ICES n 212889, and progetto di ateneo RM120172B8E7BC07. We also acknowledge the financial support by the European Union - NextGenerationEU (PRIN N. F2022NRBLPT). U.R. thanks the UKRI for financial support through the Future Leaders Fellowship MR/V025724/1 held by D. Laniel. For the purpose of open access, the authors have applied a Creative Commons Attribution (CC BY) licence to any Author Accepted Manuscript version arising from this submission.

## Author contributions

U.R., C.B., A.S. and L.E.B. designed the study. U.R., C.B., R.G., A.S. and L.E.B. conducted the experiments. U.R. and C.B. analysed the data with inputs from A.S. and L.E.B. L.J.C. and A.H. performed the theoretical calculations. U.R., A.H., A.S. and L.E.B. wrote the first draft of the paper. U.R., C.B., L.J.C., R.G., J.S.L., A.H., A.S. and L.E.B. discussed the results and finalised the manuscript. U.R., J.S.L., A.H. and L.E.B. involved in the revision of the manuscript.

## Competing interests

The authors declare no competing interests.
