## [Peer Review File · Communications Chemistry]

Structural Phase Transition in NH₄F under Extreme Pressure ConditionsReferee reports: first round

Reviewers' comments:

Reviewer #1 (Remarks to the Author):

The paper presents novel findings on the high-pressure behavior of NH_4F up to 140 GPa. Phase boundaries were determined using powder X-ray diffraction and Raman spectroscopy, and the theoretical calculations showed good agreement with the experimental data. These results are not only relevant for high-pressure studies of ammonium halides but also for understanding the high-pressure behavior of ice, given the structural similarities between H_2O and NH_4F .

While I agree with the general interpretation of the results, and have no major comments on Raman spectroscopy, or theoretical calculations, I believe the section on X-ray diffraction requires significant improvement before the paper can be considered for publication in *Communications Chemistry*. The revision should address several specific concerns regarding the diffraction part. My comments are given in the attached file.

Reviewer #2 (Remarks to the Author):

The paper combines experimental and theoretical studies to explore the phase transition of NH_4F under pressure. The synergy between experiments and theory is excellent and provides a good understanding of the vibrational-driven mechanism. The paper was written in fluent English, and both the quality of the science and the presentation are high. The following points are recommended to strengthen the conclusion:

1. The description of the point group of the Raman active modes is appreciated.
2. It might be beneficial to explore the stability of the III₁ phase while considering quantum effects. Previous studies, such as those published in *Nature* (578, 66769, 2020) and *Physical Review B* (98, 100102R, 2018), have addressed similar questions for high-pressure LaH_{10} . In particular, what is the anticipated shifting of the Raman frequencies with the effect accounted in the III₁ mode?
3. From Figure 4d, it's intriguing to observe that the phase transition follows the Bell-Evans-Polanyi principle. It's worth considering the accuracy of the kinetic barrier and whether the nudged elastic band (NEB) calculations found the correct transition states. Specifically, it raises the question of whether the transition states could be confirmed by vibrational analysis, thereby verifying the presence of saddle points. While this aspect may not be critical for the study's main purpose, it adds a layer of curiosity to the investigation.
4. It's necessary to perform a recheck of the manuscript to ensure uniform style, such as consistent formatting for figure references.

Reviewer #3 (Remarks to the Author):

I co-reviewed this manuscript with one of the reviewers who provided the listed reports. This is part of a *Communications Chemistry* initiative to facilitate training in peer review and to provide appropriate recognition for Early Career Researchers who co-review manuscripts.

The paper presents novel findings on the high-pressure behavior of NH_4F up to 140 GPa. Phase boundaries were determined using powder X-ray diffraction and Raman spectroscopy, and the theoretical calculations showed good agreement with the experimental data. These results are not only relevant for high-pressure studies of ammonium halides but also for understanding the high-pressure behavior of ice, given the structural similarities between H_2O and NH_4F .

While I agree with the general interpretation of the results, and have no major comments on Raman spectroscopy, or theoretical calculations, I believe the section on X-ray diffraction requires significant improvement before the paper can be considered for publication in *Communications Chemistry*. The revision should address several specific concerns regarding the diffraction part. My comments are in black, and the authors' text is in blue.

Major:

1. The paper lacks a phase diagram of NH_4F based on previous studies and the current work. As it stands, it is difficult for readers to follow the sequence of NH_4F phase transformations. Moreover, the phases are identified either by number or corresponding space group, which adds to the confusion.
2. The evolution of the XRD diffraction patterns in Figure 1a raises several questions:
 - a) Due to the strong crystallinity of the sample, the behavior of the small peaks is unclear and cannot be adequately evaluated. I suggest re-plotting the figure using log-units or adding inserts to show the behavior of the weak reflections. Based on the current diffraction data, it is not evident how the decision between $P-4m2$ and $P4/nmm$ phases was made. The authors should explicitly state that their conclusions are based on Raman spectroscopy and theoretical calculations, as XRD alone cannot provide a definitive answer. Were there any additional indications or evidence of a transformation to the $P4/nmm$ phase found in the XRD images?
 - b) At low 2-theta Re always appear as two reflections (1 0 0) and (1 0 1), with (1 0 1) being the strongest. The Re (1 0 1) peak shifts from 11.5° at 47.6 GPa to 12.0° at 137 GPa, consistently overlapping with the strongest peak of NH_4F . The Re line should be considered when fitting the unit cell parameters of NH_4F , especially in the pressure region below 80 GPa, where peak splitting is less pronounced. Given that the interpretation of high-pressure transitions in NH_4F heavily relies on the behavior of this peak, a discussion on the influence of the Re (101) reflection on the data interpretation should be included.
 - c) Fitting unit cell parameters using only two reflections introduces significant uncertainties. The authors mentioned that the sample was strongly preferentially oriented. Were there any spotty reflections apart from (200) and (111) on the 2D images? If so, it would be beneficial to show the cake images alongside the predicted peak positions to verify that the remaining reflections are correctly described within the proposed unit cell parameters and space group.
3. Page 8 : “The simulations of the Raman spectra therefore rely on the lattice distortion and F displacement magnitudes obtained from the diffraction experiments, with all other degrees of freedom optimised”
Please elaborate on how F-displacement magnitudes were determined from the diffraction data. Was it through Rietveld refinement or estimation from unit cell parameter distortion? The paper does not describe the structural model used for the

tetragonal phases. What Wyckoff positions do N and F occupy? A structural description should be added.

Minor:

1. P.3, Methods – Experimental: “To avoid any presence of water, the NH₄F polycrystalline samples of 99.8 % purity from Sigma Aldrich were dried at 400 K for 30 minutes before being loaded in the DACs under Argon atmosphere in a glovebox. Neon or Argon gas at a pressure of 0.2 GPa was finally loaded to serve as a pressure transmitting medium (PTM).”

Please comment on what precautions were taken to avoid water contamination during the transport of the DAC from the glovebox to the gas loading system?

2. P.4, Methods – Experimental: “The XRD data reported in the following were collected at the High-Pressure Diffraction Beamline ID15b ($\lambda = 0.4101 \text{ \AA}$) of the European Synchrotron Research Facility (ESRF) using a MAR345 image-plate detector.”

Please double-check your information about the detector used. I don't recall that ID15b (or former ID09a) has ever had a MAR345 image-plate detector.

Responses to reviewers' comments:

We give point-by-point responses below. Reviewers' comments are in color. Our responses are in black and in a different font type. In the tracked manuscript file, all changes are highlighted in color. Some minor changes were made to comply with the editorial style and formatting checklist of *Communications Chemistry*.

Reviewer #1 (Remarks to the Author):

The paper presents novel findings on the high-pressure behavior of NH₄F up to 140 GPa. Phase boundaries were determined using powder X-ray diffraction and Raman spectroscopy, and the theoretical calculations showed good agreement with the experimental data. These results are not only relevant for high-pressure studies of ammonium halides but also for understanding the high-pressure behavior of ice, given the structural similarities between H₂O and NH₄F.

*While I agree with the general interpretation of the results, and have no major comments on Raman spectroscopy, or theoretical calculations, I believe the section on X-ray diffraction requires significant improvement before the paper can be considered for publication in *Communications Chemistry*. The revision should address several specific concerns regarding the diffraction part. My comments are in black, and the authors' text is in blue.*

We thank the reviewer for their positive evaluation of our work and their valuable suggestions.

Major:

1. The paper lacks a phase diagram of NH₄F based on previous studies and the current work. As it stands, it is difficult for readers to follow the sequence of NH₄F phase transformations. Moreover, the phases are identified either by number or corresponding space group, which adds to the confusion.

We have made a figure showing a schematic phase diagram of NH₄F (now Fig. 1), and discussed it at the beginning of the Results section. The phases are now identified using their respective numbers. We hope these changes make the manuscript easier to follow.

2. The evolution of the XRD diffraction patterns in Figure 1a raises several questions:

a) Due to the strong crystallinity of the sample, the behavior of the small peaks is unclear and cannot be adequately evaluated. I suggest re-plotting the figure using log- units or adding inserts to show the behavior of the weak reflections. Based on the current diffraction data, it is not evident how the decision between P-4m₂ and P4/nmm phases was made. The authors should explicitly state that their conclusions are based on Raman spectroscopy and theoretical

calculations, as XRD alone cannot provide a definitive answer. Were there any additional indications or evidence of a transformation to the $P4/nmm$ phase found in the XRD images?

The weak peak visible at 20° at 47.6 GPa in Fig. 2b (previously Fig. 1b) is reflection (3 1 1) of $\text{NH}_4\text{F-III}$ (space group $P-4m2$). This peak shifts as expected with increasing pressure and is out of the measured angular range above 100 GPa. This is the third most intense reflection predicted for $\text{NH}_4\text{F-III}$, after (2 0 0) and (1 1 1). To support this, the simulated diffraction patterns of $\text{NH}_4\text{F-III}$ and $\text{NH}_4\text{F-III}$ at 50 GPa have been added to the Supplementary (Fig. 2).

Similarly, reflection (2 1 1) is very weak but visible in $\text{NH}_4\text{F-VIII}$ ($P4/nmm$), as can be seen in Fig. 2b. This is the third most intense reflection predicted for $\text{NH}_4\text{F-VIII}$.

Reflection (1 0 1) of $\text{NH}_4\text{F-VIII}$ is probably also visible. This is unsure because of the presence of an fcc Au peak at the same scattering angle. The peak at 10.6° in the 47 GPa pattern is indeed the main peak of the fcc Au in the sample (pressure standard), corresponding to the (1 1 1) reflection. The peak is sometimes stronger, sometimes weaker but always consistently at the expected position at all pressures. This has now been indicated in Fig. 2. The second (weaker) peak of Au, corresponding to the (2 0 0) reflection, is overlapping with the doublet from NH_4F . We collected a diffraction pattern right on the Au at every pressure point, both before and after the corresponding sample measurement, to estimate the pressure.

We understand that it can be disappointing that no more reflections are observed by XRD but even at the much lower pressure of 6 GPa, only three reflections were observed in the previous work by some of us [C. Bellin et al. *Phys. Rev. B* 96, 094110 (2017)]. This is not so much because of strong crystallinity of the sample, but simply because the diffraction intensity is concentrated on a single feature for both structures.

We now answer the question regarding the decision between the $P-4m2$ and $P4/nmm$ phases. Strictly speaking, the $P-4m2$ and $P4/nmm$ structures are identical except for the orientations of the NH_4 units, which is not detectable by XRD which is not sensitive to hydrogen atoms. In practice, we observe that phases III and VIII differ for the fact that both the tetragonal distortion and the displacement of the F atoms are bigger in the latter. But this is a minor difference, and the measured diffraction patterns change very little at the phase transition.

As the reviewer correctly noted, the conclusions are based on the combination of the XRD data with Raman spectroscopy and theoretical calculations. Using XRD, we identify the transition from phase III ($P-4m2$) to phase VIII ($P4/nmm$) based on the small jump in the c/a ratio occurring at around 115 GPa (see Fig. 2f). Raman spectroscopy also reveals a transition clearly visible in all vibrational modes at around 115 GPa (see Figs. 3 and 4). The comparison of the experimental and simulated Raman spectra unambiguously identifies this transition as from phase III to phase VIII.

We want to emphasize that, as described in the Experimental Methods section of the manuscript, we have performed multiple compression runs by both XRD and Raman spectroscopy to confirm the correspondence between the phase transitions observed with these two techniques. This approach rules out a significant effect of the compression rate and ensures that we are confidently assigning the diffraction patterns to each phase, with this assignment indicated with the color code in Fig. 2.

In the revised version of the manuscript, we have emphasized that XRD *alone* cannot provide a definitive answer.

We have also reported the simulated Raman results for the hypothetical III₁t phase with space group $P-42m$ in the Supplementary Fig. 6. This supports our choice of space group $P-4m2$ for phase III₁t. $P-42m$ and $P-4m2$ differ for the fact that only the latter allows for a displacement of the F ions. We regret not having included this information in the previously submitted version of the manuscript and hope this clarifies things.

Even though there are conclusions that cannot be drawn from our XRD data alone, we can for example discard occurrence of the two high-pressure phases of NH₄F (space groups $I4_1/amd$ and $P2_1/c$) calculated to be the more stable in a previous work published by some of us [L. J. Conway, K. Brown, J. S. Loveday, A. Hermann, *J. Chem. Phys.* 154, 204501 (2021)]. Their calculated diffraction patterns are now reported in the Supplementary Fig. 7.

Finally, the simulated Raman results for the same two high-pressure phases with space groups $I4_1/amd$ and $P2_1/c$ are now also reported in the Supplementary Figs. 8 and 9. One can easily see how the theoretical diffraction patterns and Raman spectra of these two hypothetical phases do not match our experimental results.

b) At low 2-theta Re always appear as two reflections (1 0 0) and (1 0 1), with (1 0 1) being the strongest. The Re (1 0 1) peak shifts from 11.5° at 47.6 GPa to 12.0° at 137 GPa, consistently overlapping with the strongest peak of NH₄F. The Re line should be considered when fitting the unit cell parameters of NH₄F, especially in the pressure region below 80 GPa, where peak splitting is less pronounced. Given that the interpretation of high-pressure transitions in NH₄F heavily relies on the behavior of this peak, a discussion on the influence of the Re (101) reflection on the data interpretation should be included.

Re is much less compressible than NH₄F so that the Re (1 0 1) reflection shifts from being slightly to the right of the strongest peak of NH₄F at the lowest pressures to being on the left at the highest pressure, as the reviewer correctly noted. We were aware of this and have measured diffraction patterns on the Re gasket at selected pressures. However, we think that attempting to subtract the Re signal would be nearly impossible (the position of the Re reflection is known at every pressure but its intensity is not constant and difficult to estimate) and would not affect

the conclusions. An example of pattern of the Re gasket is reported here below, together with the sample pattern at the same pressure of 73.7 GPa. The intensity of the Re pattern was arbitrarily scaled so that the Re (1 0 0) reflection has about the same magnitude in the two patterns.

One can appreciate how the Re (1 0 1) reflection is much weaker than the overlapping NH_4F signal. This is also Supplementary Fig. 3.

At pressures above 80 GPa, when the splitting is more pronounced, the two-Gaussians fit to the data is also good, proving that the influence of the Re (1 0 1) reflection is minimal. There is no reason to believe that the intensity of the Re peaks would decrease with pressure and consistently the intensity of the Re (1 0 0) reflection does decrease with pressure (see Fig. 2b). Below 80 GPa, the splitting is less pronounced and the Re peak might slightly affect the results for the lattice parameters. In the revised version of the manuscript, we have added error bars to Figs. 2d, 2e and 2f to account for this.

Finally, we would like to note that: first, the lattice parameters of NH_4F from 47 to 80 GPa are not the main outcome of this paper and the obtained volumes are not different from those expected by simulations (see Fig. 2d) and second, the reviewer will agree that it is always challenging to model a small splitting in high-pressure data, regardless of an overlapping (for example) Re signal. Significant peak broadening is inevitable in high-pressure experiments and, with the tetragonal distortion being below 4% at pressures below 80 GPa, this represents the main (intrinsic) limitation of this part of the X-ray diffraction data analysis. Nonetheless, looking at Fig. 2c, there

is no doubt that the most reasonable interpretation of the available data is a progressive splitting of the main NH_4F peak with pressure.

c) Fitting unit cell parameters using only two reflections introduces significant uncertainties. The authors mentioned that the sample was strongly preferentially oriented. Were there any spotty reflections apart from (200) and (111) on the 2D images? If so, it would be beneficial to show the cake images alongside the predicted peak positions to verify that the remaining reflections are correctly described within the proposed unit cell parameters and space group.

An example of cake image (azimuth as a function of 2θ) and the corresponding integrated 1D pattern without any background subtraction are reported here below. This is at 80.9 GPa.

As we discuss in the manuscript, the intensity is concentrated on a single feature (11.9° at this pressure) and there are not so many other visible reflections. We feel that there is not much more to learn from such cake images.

Regarding the question on whether the remaining reflections are correctly described within the proposed unit cell parameters and space group, we feel we have already answered in point a). We can complement that with two further points: first, a NH_4F phase having space group $P4/nmm$ emerged from the previously published as well as our new computer simulations, independently of the XRD data. $P4/nmm$ is also a known structure of other ammonium halides. Second, $P-4m2$ is the space group directly obtained from $P4/nmm$ after rotating every second NH_4 unit so that they are all aligned as in phase III. In parallel to this, starting from phase III of NH_4F (space group $P-43m$) and allowing for a tetragonal distortion plus a displacement of the F ions, $P-4m2$ emerges as the space group with the highest possible symmetry.

3. Page 8 : “The simulations of the Raman spectra therefore rely on the lattice distortion and F displacement magnitudes obtained from the diffraction experiments, with all other degrees of freedom optimised”

Please elaborate on how F-displacement magnitudes were determined from the diffraction data. Was it through Rietveld refinement or estimation from unit cell parameter distortion? The paper does not describe the structural model used for the tetragonal phases. What Wyckoff positions do N and F occupy? A structural description should be added.

The F-displacement magnitudes of phase III_t were determined by comparing experimental and simulated Raman spectra. Specifically, we used the experimental unit cell at 114 GPa, set up a series of 30 F-displacements of varying amplitudes, optimised all other degrees of freedom, and then simulated the Raman spectra. We compared the series of simulated spectra to the experimental spectrum, which allowed us to establish the displacement amplitude at 114 GPa, where the F atoms are at position (0, 1/2, z) with $z=0.0405$. We then assumed a linear pressure dependence of the displacement on the c/a ratio, starting from $z=0$ at 40 GPa, the onset of phase III_t, and finally we used the obtained interpolation for the simulations of the Raman spectra of phase III_t at various pressures (reported in Fig. 4 of the main text). We have added the above description to the revised manuscript and two new figures to the Supplementary (Figs. 4 and 5).

Concerning the second part of the comment, the DFT-optimised structural models of all the phases discussed in this manuscript are now summarized in the Supplementary Tables 1 and 2. Crystallographic information files are also provided. We apologize for not having provided all details in the previously submitted version of the manuscript.

Minor:

1. P.3, Methods – Experimental: “To avoid any presence of water, the NH₄F polycrystalline samples of 99.8 % purity from Sigma Aldrich were dried at 400 K for 30 minutes before being loaded in the DACs under Argon atmosphere in a glovebox. Neon or Argon gas at a pressure of 0.2 GPa was finally loaded to serve as a pressure transmitting medium (PTM).”

Please comment on what precautions were taken to avoid water contamination during the transport of the DAC from the glovebox to the gas loading system?

Following a well-established procedure, the DAC had been closed in the glovebox in order to avoid water contamination during transport, meaning that the sample chamber was sealed but there was no pressure on the sample. This is easy to do when the sample does not fill the sample chamber completely. Then when the DAC was inside the gas loading system and the (Neon or Argon) gas pressure was 0.2 GPa, the DAC was opened to let the gas diffuse into the sample chamber (the gas loading system provides motorized access to the screws of the DAC). Finally the DAC was closed inside the gas loading system to increase the pressure and trap the gas.

In the revised manuscript, examples of photomicrographs of the three loaded DACs are reported in the Supplementary Fig. 1.

2. P.4, Methods – Experimental: “The XRD data reported in the following were collected at the High-Pressure Diffraction Beamline ID15b ($\lambda = 0.4101 \text{ \AA}$) of the European Synchrotron Research Facility (ESRF) using a MAR345 image-plate detector.”

Please double-check your information about the detector used. I don't recall that ID15b (or former ID09a) has ever had a MAR345 image-plate detector.

We have double-checked and ID15b had a MAR345 image-plate detector. However, the latter had been replaced shortly before our experiment with a better detector, namely an Eiger2 X 9M CdTe. Obviously, we apologize for this inaccuracy and have now amended the text of the manuscript. We thank the reviewer for their attention.

Reviewer #2 (Remarks to the Author):

The paper combines experimental and theoretical studies to explore the phase transition of NH₄F under pressure. The synergy between experiments and theory is excellent and provides a good understanding of the vibrational-driven mechanism. The paper was written in fluent English, and both the quality of the science and the presentation are high. The following points are recommended to strengthen the conclusion:

We thank the reviewer for their positive evaluation of our work and their valuable suggestions.

1. The description of the point group of the Raman active modes is appreciated.

We have added tables to the Supplementary with the mode symmetries of the Raman-active modes for phases III, III_t, and VIII at a representative pressure (Supplementary Tables 3, 4, and 5). We refer to the tables in the revised manuscript.

2. It might be beneficial to explore the stability of the III_t phase while considering quantum effects. Previous studies, such as those published in Nature (578, 66-69, 2020) and Physical Review B (98, 100102R, 2018), have addressed similar questions for high-pressure LaH₁₀. In particular, what is the anticipated shifting of the Raman frequencies with the effect accounted in the III_t mode?

Nuclear quantum effects can stabilise crystal structures that are found to be dynamically unstable if nuclei are assumed to be classical point particles. The works cited by the reviewer are exemplary in that area: they show the stabilisation of a high-symmetry structure (cubic LaH₁₀ in these cases), driven by symmetrisation of the hydrogenic sublattice. The III_t phase is quite different in that regard: it is a *low-symmetry* structure, driven by displacement of the *fluorine* sublattice (hydrogens seem to remain in place as in phase III). From a physical standpoint it is hard to see how nuclear quantum effects could play a role for the fluorine nuclei, while from a technical standpoint the low-symmetry structure of III_t makes for an ill-defined starting point for the type of self-consistent phonon calculations done e.g. in the 2020 *Nature* paper by Errea *et al.*

Nuclear quantum effects for hydrogen could result in NH₄ cations being quantum or (at elevated temperature) classical rotors. That implies that NH₄ cations, upon compression, evolve from fixed orientation (in phase III) to plastic/rotational states (in phase III_t) and back to fixed orientation (in phase VIII). There is no indication for this in the experimental data, and no precedence in the literature either: plastic/rotational states in water, ammonia, or their mixtures, are all arrived at upon heating, not compressing.

On balance we feel it unlikely that nuclear quantum effects can explain the emergence of phase III_t, and we have included the above arguments in the Discussion section of the revised manuscript (page 13).

3. From Figure 4d, it's intriguing to observe that the phase transition follows the Bell-Evans-Polanyi principle. It's worth considering the accuracy of the kinetic barrier and whether the nudged elastic band (NEB) calculations found the correct transition states. Specifically, it raises the question of whether the transition states could be confirmed by vibrational analysis, thereby verifying the presence of saddle points.

While this aspect may not be critical for the study's main purpose, it adds a layer of curiosity to the investigation.

We thank the reviewer for those suggestions. We agree that the barriers we report are tied to the assumed reaction coordinate, which in our NEB calculations will remain close to the F displacement and NH₄ rotation. We have calculated the Raman-active modes at the saddle points at 50/100/150 GPa, and reported them in the Supplementary together with the structural information (Supplementary Figs. 10 and 11, and Supplementary Table 6). At all pressures the transition states feature much higher frequency N—H stretch modes, which are not seen in experiments. However, we have no information about the lifetime of these (or any other) transition states, so this does not rule out our transition paths.

We have added the above information and a brief discussion of the BEP principle to the revised manuscript (page 12).

4. It's necessary to perform a recheck of the manuscript to ensure uniform style, such as consistent formatting for figure references.

We have checked the manuscript for both scientific and stylistic inaccuracies. Changes were made where necessary.

Specifically, figure references are now formatted uniformly. We have also, for example, uniformized information given in the various subsections of the Results section: the transition pressure into phase VIII is now given to be around 115 GPa everywhere (rather than values ranging between 114 and 120 GPa as in the previous version of the manuscript). We have deleted some misleading text from the caption of Fig. 4. We have followed the editorial formatting guide for other minor stylistic changes.

Reviewer #3 (Remarks to the Author):

I co-reviewed this manuscript with one of the reviewers who provided the listed reports. This is part of a Communications Chemistry initiative to facilitate training in peer review and to provide appropriate recognition for Early Career Researchers who co-review manuscripts.

We thank the reviewer for their work.

Decision letter and referee reports: second round

REVIEWERS' COMMENTS:

Reviewer #1 (Remarks to the Author):

I would like to express my appreciation for the effort the authors have put into addressing the comments and suggestions. The revisions have significantly improved the quality and clarity of the manuscript. I have carefully reviewed the updated version and am pleased to report that all my concerns have been thoroughly addressed.

I have no further comments or suggestions for improvement. I am satisfied with the current version and recommend the paper for publication.

Reviewer #2 (Remarks to the Author):

The revision is acceptable, and the manuscript is recommended for publication.

Reviewer #3 (Remarks to the Author):

I co-reviewed this manuscript with one of the reviewers who provided the listed reports. This is part of the Communications Chemistry initiative to facilitate training in peer review and to provide appropriate recognition for Early Career Researchers who co-review manuscripts.

Reviewer #1 (Remarks to the Author):

I would like to express my appreciation for the effort the authors have put into addressing the comments and suggestions. The revisions have significantly improved the quality and clarity of the manuscript. I have carefully reviewed the updated version and am pleased to report that all my concerns have been thoroughly addressed.

I have no further comments or suggestions for improvement. I am satisfied with the current version and recommend the paper for publication.

Reviewer #2 (Remarks to the Author):

The revision is acceptable, and the manuscript is recommended for publication.

Reviewer #3 (Remarks to the Author):

I co-reviewed this manuscript with one of the reviewers who provided the listed reports. This is part of the Communications Chemistry initiative to facilitate training in peer review and to provide appropriate recognition for Early Career Researchers who co-review manuscripts.

We thank all three reviewers for their work and their positive evaluation of our manuscript.